# Effectiveness of Equine-Assisted Intervention as a Therapeutic Strategy for Improving Adaptive Behaviour in Children with Autism Spectrum Disorder

**DOI:** 10.3390/healthcare13162014

**Published:** 2025-08-15

**Authors:** Carmen María Martínez Moreno, José Manuel Hernández Garre, Paloma Echevarría Pérez, Isabel Morales Moreno, Eva Vegue Parra, Eloína Valero Merlos

**Affiliations:** Department of Social and Healthcare Research and Development, Catholic University of Murcia, 30107 Murcia, Spain; cmm97b@gmail.com (C.M.M.M.); pechevarria@ucam.edu (P.E.P.); imorales@ucam.edu (I.M.M.); evegue@ucam.edu (E.V.P.); evalero@ucam.edu (E.V.M.)

**Keywords:** autism spectrum disorder, equine assisted intervention, adaptive behavior, social skills, communication, daily living skills, motor skills

## Abstract

**Background/Objectives**: This study examines the effectiveness of equine-assisted intervention (EAI) in improving adaptive behaviour and motor skills in children with autism spectrum disorder (ASD). **Methods**: To that effect, a self-controlled experimental analytical study has been designed, which is longitudinal and prospective in nature, with pre- and post-intervention measures, using the Vineland Adaptive Behavior Scale II (VABS-II) as the assessment instrument. The sample consists of 19 children who participated in weekly therapeutic sessions involving horses for eight months; these sessions included horseback riding, groundwork, hygiene, and preparation of the horse. **Results**: The results show significant improvements both in the overall score of the VABS-II test ($\bar{x}$pre: 65.84 ± 10.38–$\bar{x}$post: 72.47 ± 16.21, *p* = 0.003) and in the areas of communication ($\bar{x}$pre: 64.84 ± 15.50 ~ $\bar{x}$post: 72.26 ± 21.93, *p* = 0.010), social skills ($\bar{x}$pre: 61.26 ± 8.99 ~ $\bar{x}$post: 66.53 ± 13.79, *p* = 0.008) and daily living skills (DLS) ($\bar{x}$pre: 66.21 ± 11.15 ~ $\bar{x}$post: 69.95 ± 12.32, *p* = 0.0004), as well as a non-significant slight improvement in motor skills ($\bar{x}$pre: 72.50 ± 8.83 ~ $\bar{x}$post: 75.17 ± 7.88, *p* = 0.363). In addition, these gains were greater in those children attending standard classroom settings and receiving early stimulation. **Conclusions**: This study suggests equine-assisted intervention (EAI) may contribute to improvements in adaptive behaviour, including communication, social skills, and daily living skills, in children with autism spectrum disorder (ASD). Benefits were notably enhanced in children receiving early stimulation within standard classroom settings.

## 1. Introduction

ASD represents a varied group of neurodevelopmental disorders characterised by repetitive or restricted behavioural patterns and persistent deficits in communication and social interaction [1]. Due to the symptoms, severity, and negative impact of the disorder, early diagnosis and intervention are necessary to help the child and their family adapt. The early approach is based on interdisciplinary collaboration among different professionals in the healthcare sector, in order to achieve improvements in adaptive, cognitive, and linguistic processes [2]. The fundamental objectives of treatment consist of maximising patients’ functional independence and quality of life, minimising the core characteristics of the disorder, encouraging social skills, reducing disruptive behaviour, and educating families. Various therapeutic approaches exist, among which we distinguish pharmacological treatment—aimed at alleviating secondary symptoms such as anxiety, hyperactivity, aggressiveness, and stereotyped behaviour [3]—as well as behavioural interventions such as the Lovaas method and the Early Start Denver Model [4,5].

However, despite the large number of existing therapeutic approaches, current therapies do not yet have a sufficiently established scientific basis. Moreover, the prevalence of the disorder has continued to rise in recent years, creating a negative impact on the lives of those affected [6]. Therefore, the need for more extensive research has become increasingly urgent in order to assess the clinical effectiveness of complementary or unconventional interventions. This is an area where a wide variety of alternative therapies has expanded, such as animal-assisted interventions (AAIs), which appear to offer benefits at various levels [7]. Among these therapies, equine-assisted intervention (EAI) has gained increasing attention and is mainly applied in the treatment of individuals with ASD [8]. It is a physical, psychological, educational, social, and occupational intervention method that employs an adequately trained horse as the therapeutic medium, with the ultimate goal of improving the quality of life of individuals with special needs. It represents a holistic form of therapy encompassing not only physiotherapy-related activities, but also neuromotor, sensory, emotional, cognitive, and social development. Improvements at the physical level facilitate experimentation and exploration of the environment, which in turn supports the development of cognitive and social abilities [9]. Participating in these sessions in a natural setting is highly engaging for children, as nature helps them forget they are undergoing therapy—unlike conventional hospital-based sessions, which often lack sufficient motivational stimuli [10].

There is currently an increasing number of publications that investigate the effectiveness of EAI in individuals with ASD, demonstrating progress in various functional areas. In the area of motor skills, studies have reported improvements in posture, walking, and coordination [11,12,13,14,15]; in the area of social skills, other studies have identified enhancements in sensory stimulation, concentration, and memory; and in the area of communication, research has shown developments in language acquisition and social integration [16,17,18,19,20,21,22,23,24,25,26]. Ultimately, these studies have evaluated the potential improvements that EAI may bring in terms of adaptive behaviour, including aspects such as emotional stability, reductions in fear, increased initiative, self-esteem, self-confidence, communication, and social skills [27,28].

In recent years, this line of research has been strengthened by more robust studies and systematic reviews published in 2023 and 2024, further supporting the efficacy of equine-assisted interventions for individuals with ASD. A comprehensive systematic review highlighted significant improvements in motor outcomes such as coordination, balance, posture, and motor performance following EAI in autistic children [29]. Similarly, a meta-analysis demonstrated consistent benefits in social communication and social cognition, with positive outcomes measured through validated tools such as the Social Responsiveness Scale (SRS) [30]. Additionally, a recent longitudinal study involving 86 children with ASD (levels 1 to 3) reported substantial gains in daily living skills and socialisation following a structured EAI programme of 20 weekly sessions [31]. These findings reaffirm the growing body of evidence supporting EAI as a promising complementary therapeutic strategy across multiple functional domains.

Nevertheless, many of those previous research efforts have certain limitations due to their observational nature, small sample sizes, and very short exposure periods; due to this, the real efficacy of those interventions remains uncertain. Our study addresses a notable gap by implementing a longer exposure period (8 months), a regional (Spanish) cultural context, and an age-diverse cohort of children with moderate-to-high symptom severity, providing preliminary longitudinal insight from a real-world setting. Thus, it seems appropriate to continue to explore the phenomenon in question, focusing on its possible benefits in terms of social, communication, and motor skills, as well as in the area of DLS. To that effect, our study involves a larger sample and a longer exposure period than most other studies and incorporates the Vineland Adaptive Behaviour Scales-II (VABS-II) as a measurement tool. Some of the sources consulted [11,16,27,28] report significant changes in certain domains and subdomains assessed with this scale; however, there is insufficient evidence regarding the efficacy of the intervention for other related variables. Ultimately, this research contributes to a broader exploration of the potential impact of EAI in addressing ASD, whereby such potential is assessed in terms of various independent variables, namely gender, type of complementary therapy, type of classroom setting, attendance or non-attendance at early intervention, and age. Nevertheless, it is important to note that this study is intended as an exploratory pilot investigation rather than a definitive efficacy trial. Given the absence of a control group and the use of a pre-post design, causal attributions must be made with caution. The results should be interpreted as preliminary indications of potential benefits, rather than conclusive proof of therapeutic efficacy. This framing reinforces the scientific integrity of the study while highlighting its value in guiding future controlled trials.

## 2. Materials and Methods

We used a quantitative method involving a self-controlled experimental analytical study (pre- and post-intervention measures), which was longitudinal and prospective in nature, based on an equine-assisted intervention (EAI) programme for children with autism spectrum disorders (ASD).

Given the ethical and logistical constraints associated with access to this type of therapy, and the limited regional availability of structured EAI programmes, the present study was designed as an exploratory, quasi-experimental investigation without a control group. Due to practical and ethical considerations—including the limited local access to EAI and the impossibility of withholding potentially beneficial therapies from children with ASD—it was not feasible to include a matched control group or to randomly assign participants. Although the absence of a matched control group limits the ability to isolate the effects of EAI from potential confounding variables (e.g., natural developmental progression or concurrent therapies), this design allows for a preliminary evaluation of potential therapeutic benefits. The results obtained are intended to inform future studies incorporating randomised controlled trials or alternative designs such as propensity score matching to account for confounding factors.

### 2.1. Participants

There were 19 participants—4 girls and 15 boys between 6 and 18 years old. All domains were evaluated in all children except for the motor skills domain of the VABS-II, where only 6 children could participate (n = 6), due to this scale’s restriction that specifies that only children under 6 years of age can be assessed. The children were enrolled in conventional therapies with medical follow-up and mandatory school attendance (primary or secondary education). In addition, some participants in our sample were receiving psychological support, speech therapy, or early intervention (physiotherapy and sensorimotor stimulation). To provide further detail on the concomitant therapies and support received, we can specify the frequency of these sessions. The psychological support sessions were delivered on a weekly basis, while both speech therapy and early intervention (physiotherapy and sensorimotor stimulation) were conducted three times per week. All of these sessions had a typical duration of one hour. Regarding educational placement, 11 children were enrolled in a standard classroom setting—referring to regular educational environments with no intensive support needs or with milder special educational needs (SEN) profiles—while 8 children attended an “aula abierta” (open classroom). These open classrooms, also known as specialised support classrooms within mainstream schools, are designed for students with more severe or complex SEN profiles. They offer highly individualised curricula, continuous specialised support (such as therapeutic pedagogy, educational assistants, speech therapists, and sometimes physiotherapists), and aim to balance intensive intervention with integration in general school life. These classrooms promote socialisation by enabling students to participate in certain activities and subjects (e.g., physical education, music, recess, or extracurricular activities) alongside their peers in mainstream classrooms.

The children were selected from the EAI integrated care centres of the Tiovivo Foundation in Murcia and Cartagena, Spain. Each child included in the study had an initial ASD diagnosis meeting the criteria of the DSM-IV-TR and ICD-10. Regarding symptom severity, the children presented a range of clinical profiles, from non-verbal children with no expressive language to children with limited but functional verbal skills without fluent speech. According to the Autism Diagnostic Observation Schedule—Second Edition (ADOS-2), the participants presented total scores ranging from 8 to 15, indicating moderate to high symptom severity across the sample. The inclusion criteria were a diagnosis of ASD, 6–18 years of age, first-time participation in EAI, and written informed consent signed by the parents as the child’s legal guardians. As for the exclusion criteria, these were severe motor or neurological impairments, a fearful response towards horses, refusal to participate in the study, and prior involvement in EAI.

The protocol of the study was approved by the Tiovivo Foundation. This approval was granted in compliance with national and regional legislation regarding animal welfare, specifically Spanish Law 7/2023 on Animal Protection and Welfare, the regional Animal Protection Law of the Region of Murcia, and Royal Decree 53/2013 concerning the ethical evaluation of procedures involving animals. The protocol adhered to best practices in equine welfare and was reviewed and authorised by the foundation’s ethics committee to ensure proper care, handling, and ethical use of horses within the therapeutic context.

The sampling method used was one of convenience, owing to the low prevalence of children with autism spectrum disorder (ASD) undergoing equine-assisted intervention (EAI) within our region. Consequently, a formal sample size calculation or power analysis was not feasible. Furthermore, all participants were recruited from the Tiovivo Foundation, the only institution in the area offering this type of structured EAI therapy with the necessary clinical and educational coordination. Although this limits the generalisability of the findings, the centre’s unique expertise and infrastructure enabled consistent application of the intervention across all cases.

### 2.2. Setting

The EAI sessions were carried out at the two specialised centres of the Tiovivo Foundation in Murcia (n = 10) and Cartagena (n = 9). The therapists who conducted the sessions were the same in both centres; they were professionals from either the healthcare or education sectors, with specialisation in therapeutic horseback riding and over five years of experience in this field. To ensure consistency across EAI sessions, prior meetings were held between the principal investigator and the therapists in order to define the structure, frequency, and duration of the sessions. The setting included riding trails and riding schools, stables, tack rooms, and multi-purpose rooms for the workshops.

### 2.3. EAI Sessions

EAI sessions were conducted individually for each child on a weekly basis over a period of eight months, resulting in a total of 32 sessions per child. As shown in Table 1, each session lasted approximately 45–60 min and was divided into several parts: five minutes for initial contact, 15 min for preparation of the child and the horse, 30 min of horseback riding, and 10 to 15 min for closing the session. The main objective was for children to begin establishing visual contact with both the horse and the therapist. Another aim was for children to demonstrate signs of comprehension in carrying out simple tasks, stabilise their mood, and reduce motor instability.

### 2.4. Measurements

Each subject in the study was evaluated 15 days prior to the beginning of the EAI sessions (pre-intervention) and 15 days after completing them (post-intervention), using the Vineland Adaptive Behavior Scale-II as the measurement tool (VABS-II, Sparrow et al., 2005 [32]). VABS-II is a validated and standardised instrument that evaluates adaptive behaviour among other functions. It entails a semi-structured interview conducted with parents and therapists, and is divided into four domains, which in turn are subdivided into subdomains of communication (receptive, expressive, and written), DLS (personal, domestic, community), social skills (interpersonal relationships, play and leisure time, and coping skills), and motor skills (fine and gross motor skills). It includes a three-point Likert-type scale (P0 = the behaviour never occurs; P1 = it occurs sometimes; P2 = it always occurs), and two additional options (DK = do not know; N/O = no opportunity). It comprises a total of 383 items to be evaluated—99 in the communication domain, 109 in the DLS domain, 99 in the social skills domain, and 76 in the motor skills domain. The scale defines five levels of adaptive behaviour based on the total score from the domains and subdomains, namely the low adaptive level from 20 to 70 points, moderately low from 71 to 85 points, adequate level from 86 to 114 points, moderately high level from 115 to 129 points, and high level from 130 to 160 points.

### 2.5. Statistical Analysis

To analyse the EAI effectiveness level through pre- and post-intervention measurements, a bivariate analysis was performed by means of a Student’s *t*-test for dependent samples, once it was confirmed that the variables followed a normal distribution verified by means of the Shapiro–Wilk test (n < 50). In order to verify whether or not there was a relationship between the effectiveness of EAI and the categorical independent variables, a bivariate analysis was carried out using a Student’s *t*-test for independent variables in case the independent variables were dichotomous, and an analysis of variance (ANOVA) was performed when the independent variables had more than two categories. The intergroup analysis was carried out both on the basis of the post-intervention overall score of the VABS-II test and on the basis of score increases pre- and post-intervention. All of this was performed after using the Shapiro–Wilk normality test (n < 50) to confirm that the quantitative analysis variables followed a normal distribution, as well as using the Levene test—or Welch’s *t*-test when the Levene test could not be used—to confirm that there was homogeneity among variances (homoscedasticity). Finally, to determine whether or not there was an association between the effectiveness of the intervention and the quantitative independent variables, Pearson’s correlation coefficient was used. Given the exploratory nature of this pilot study, a decision was made not to apply a formal correction for multiple comparisons. The numerous statistical tests conducted on the Vineland Adaptive Behavior Scales are not independent but rather analyse a hierarchical structure of related constructs. A conservative correction method, such as Bonferroni, could obscure genuine yet modest effects that warrant further investigation. Therefore, we prioritised the detection of potential effects and the generation of hypotheses, acknowledging that the findings should be interpreted with caution as preliminary results to be confirmed in future, more rigorous studies. All of the statistical processes were carried out in SPSS 21.0. In the hypothesis testing, the results considered significant were those obtained for a value of *p* < 0.05.

### 2.6. Ethical Considerations

The study was conducted in accordance with the guidelines of the Declaration of Helsinki. Written informed consent was obtained from all parents or legal guardians of the children with autism spectrum disorder (ASD) who participated in the study. Furthermore, ethical approval was granted by the Ethics Committee of the Tiovivo Foundation, Region of Murcia, on 18 June 2015 (Protocol Code: 8-tiv).

## 3. Results

As can be observed in Table 2, following EAI there is a significant increase in the overall score averages of the VABS-II, where the subjects of the study progress from a low adaptive level to a moderately low level (Pglobal $\bar{x}$pre: 65.84 ± 10.38 ~ $\bar{x}$post: 72.47 ± 16.21; t(18): −3.39, *p* = 0.003).

Similarly, it can also be observed that EAI was followed by an increase in VABS-II score averages in the four domains, showing the existence of statistically significant associations in the areas of communication, DLS, and social skills, but not in the area of motor skills (Dcom $\bar{x}$pre: 64.84 ± 14.50 ~ $\bar{x}$post: 72.26 ± 21.93; t(18): −2.88, *p* = 0.010; Ddls $\bar{x}$pre: 66.21 ± 11.15 ~ $\bar{x}$post: 69.95 ± 12.32; t(18):−4.30, *p* = 0.0004; Dsoc $\bar{x}$pre: 61.26 ± 8.99 ~ $\bar{x}$post: 66.53 ± 13.79; t(18): −2.97, *p* = 0.008; Dms $\bar{x}$pre: 72.50 ± 8.83 ~ $\bar{x}$post: 75.17 ± 7.88; t(18): −1.00, *p* = 0.363).

With respect to subdomains, one can observe statistically significant increases in the subdomains of receptive, expressive, and written communication (Srec $\bar{x}$pre: 9.11 ± 2.40 ~ $\bar{x}$post: 10.58 ± 3.32; t(18): −3.03, *p* = 0.007; Sexp $\bar{x}$pre: 7.53 ± 2.98 ~ $\bar{x}$post: 8.58 ± 4.07; t(18): −2.34, *p* = 0.031; Swri $\bar{x}$pre: 9.84 ± 4.76 ~ $\bar{x}$post: 11 ± 5.53; t(18): −2.18, *p* = 0.043), in the subdomains of personal, domestic, and community DLS (Sper $\bar{x}$pre: 9.05 ± 3.51 ~ $\bar{x}$post: 9.95 ± 4; t(18): −3.25, *p* = 0.004; Sdom $\bar{x}$pre: 10.05 ± 1.54 ~ $\bar{x}$post: 10.68 ± 1.56; t(18): −2.72, *p* = 0.014; Scom $\bar{x}$pre: 8.21 ± 3.02 ~ $\bar{x}$post: 9.37 ± 2.96; t(18): −2.13, *p* = 0.047) and in the social skills subdomains that involve interpersonal relationships and play and leisure time (Sir $\bar{x}$pre: 7.68 ± 2.02 ~ $\bar{x}$post: 8.68 ± 2.68; t(18): −3.00, *p* = 0.008; Spl $\bar{x}$pre: 7.32 ± 2.49 ~ $\bar{x}$post: 8.47 ± 3.53; t(18): −2.35, *p* = 0.030). However, no statistically significant improvements can be observed in the social skills subdomain that involves coping skills (Scs $\bar{x}$pre: 9.05 ± 1.31 ~ $\bar{x}$post: 9.63 ± 2.31; t(18): −1.50, *p* = 0.150), or in the gross and fine motor skills subdomains (Sgr $\bar{x}$pre: 11.67 ± 4.22 ~ $\bar{x}$post: 12.00 ± 3.74; t(18): −1.00; *p* = 0.363; Sfi $\bar{x}$pre: 12.33 ± 5.04 ~ $\bar{x}$post: 13.00 ± 4.42; t(18): −1.00, *p* = 0.363) (Figure 1). It is important to note that the motor skills domain of the VABS-II was assessed in only six participants due to the test’s age restriction (applicable exclusively to children under six years old). As a result, the statistical power of this analysis is limited, and the findings should be interpreted with caution and in an exploratory manner. Although the results are not statistically significant, a slight trend towards improvement was observed in both subdomains.

To add transparency and provide preliminary insight into potential effects, exploratory comparisons were performed on key subgroups. These findings, presented in Table 3, should be interpreted with caution due to the study’s design and small sample size. They are intended to highlight potential trends for future research rather than to provide definitive proof of efficacy across specific subgroups.

As can be observed in Table 3, the children in a standard classroom setting had higher scores in the VABS-II compared to those in an open classroom setting; this is a significant finding that is supported by the overall score averages post-intervention ($\bar{x}$standard-classroom: 79.09 ± 12.21 ~ $\bar{x}$open-classroom: 64.50 ± 11.37; t(17): 2.64, *p* = 0.017), as well as by the average increases in score averages post-intervention ($\bar{x}$standard-classroom: 10.64 ± 11.49 ~ $\bar{x}$open-classroom: 2.63 ± 2.61; t(11.392): 2.23, *p* = 0.046). Similarly one can observe higher scores also in children that receive early attention compared to those who do not; this observation is verified both by the post-intervention overall score averages ($\bar{x}$YES-ATT: 80.43 ± 15.98 ~ $\bar{x}$NO-ATT: 66.75 ± 10.98; t(17): 2.21, *p* = 0.041) and by the post-intervention average increases in the scores ($\bar{x}$YES-ATT: 13.43 ± 12.43 ~ $\bar{x}$NO-ATT: 1.83 ± 2.29; t(6.238): 2.44, *p* = 0.049). However, no statistically significant differences can be observed with respect to the gender variable either in the post-intervention overall score averages ($\bar{x}$boys: 73.33 ± 18.13 ~ $\bar{x}$girls: 69.25 ± 4.49; t(17): 0.43, *p* = 0.667) or in the post-intervention increases ($\bar{x}$boys: 7.27 ± 9.47 ~ $\bar{x}$girls: 4.25 ± 2.50; t(17): 0.61, *p* = 0.544). Moreover, the results do not show statistically significant differences in post-intervention overall scores (*p* = 0.849) or in the post-intervention average increases (*p* = 0.280) with respect to the age of the children. Similarly, the complementary therapy variable also does not yield statistically significant results, as the post-intervention overall score averages are higher in the group that received psychological support, followed by the group that did not undergo any type of therapy, and finally by those who received speech therapy ($\bar{x}$None: 66.89 ± 10.84 ~ $\bar{x}$Psych: 73.25 ± 8.95 ~ $\bar{x}$Speech: 64 ± 12.97; t(18): 0.27, *p* = 0.762). As to the average increase in the scores post-intervention, the greatest increase can be observed in the group of children that did not receive any type of complementary therapy, followed by those who received psychological support, and finally those who participated in speech therapy sessions ($\bar{x}$None: 8 ± 12.09 ~ $\bar{x}$Psych: 7 ± 3.16 ~ $\bar{x}$Speech: 4.33 ± 3.32; t(18): 0.31, *p* = 0.736) (Figure 2).

## 4. Discussion

In line with prior research [27,28], the results of this study suggest the potential role of EAI as a therapeutic strategy for children with ASD, as they indicate a statistically significant improvement in adaptive behaviour after 8 months of exposure to the therapy. In an analysis by domain, a clear improvement was observed in communication, social skills, and DLS, as well as a slight improvement in motor skills without statistical significance.

If we contrast this study with prior evidence, we find that there is a large number of research efforts that associate EAI with improvement in social, behavioural, and communication skills [16,17,18,19,20,21,22,23,24,25,26,29], including two studies that use the VABS as a measurement tool [27,28]. The study by Borgi et al. [27] uses the VABS (1984), coinciding with our study in that it finds significant improvements in both the communication domain (Dcom: *p* = 0.006) and in the social skills domain (Dsoc: *p* = 0.003), although the results of that study are limited as they do not specify results by subdomain. The study conducted by Azjeman et al. [28], on the other hand, used the same scale as that in our study, i.e., VABS-II [32], also conducting an analysis by subdomain. In this context there are certain similarities and discrepancies; on one hand, both studies show statistically significant improvement in the communication domain (Azjeman et al. [28]: Dcom: *p* = 0.042; this study: Dcom: *p* = 0.010) and the social skills domain (Azjeman et al. [28]: Dsoc: *p* = 0.027; this study: Dsoc: *p* = 0.008), while on the other hand, there are discrepancies with regard to subdomains:. While the study by Azjeman et al. [28] only establishes a significant improvement in the communication domain and the receptive communication subdomain, it does not show significant improvement either in the expressive or in the written communication subdomain (Srec: *p* = 0.026; Sexp: *p* = 0.157; Swri: *p* = 0.059); our study, however, suggests improvements in all three subdomains (Srec: *p* = 0.007; Sexp: *p* = 0.031; Swri: *p* = 0.043). With regard to the area of social skills, the study by Azjeman et al. [28] does not show an improvement in the subdomains of interpersonal relationships or play and leisure time, yet it does show improvement in the coping skills subdomain (Sir: *p* = 0.197; Spl: *p* = 0.109; Scs: *p* = 0.038). In our study, paradoxically, we noted the opposite results; we identified significant improvements in the subdomains of interpersonal relationships and play and leisure time but not in the coping skills subdomain (Sir: *p* = 0.008; Spl: *p* = 0.030; Scs: *p* = 0.150). It must be pointed out that in the study conducted by Azjeman et al. [28], the sample consisted of only six children and the time of exposure was three months as opposed to 19 children and eight months of exposure in our study. These individual study results are further supported by a recent meta-analysis by Xiao et al. [30], which reviewed equine-assisted interventions in ASD populations and reported consistent improvements in social communication, social cognition, and linguistic skills. This systematic evidence reinforces the notion that EAI can positively impact the very domains evaluated in our study, lending additional support to our findings within a broader empirical context.

With respect to the DLS area, the results of our study coincide with those of Borgi et al. [27], which suggest a significant improvement in this area (Ddls: *p* = 0.001), although, as already mentioned, the research does not include an analysis by subdomain. However, the results of our research differ from those of the study conducted by Azjeman et al. [28], where no statistically significant association was reported with respect to DLS either in the domain or in the subdomains (Ddls: *p* = 0.093; Sper: *p* = 0.131; Sdom: *p* = 0.334; Scom: *p* = 0.102), as opposed to our study, which indicates this association both in the domain and in the subdomains (Ddls: *p* = 0.0004; Sper: *p* = 0.004; Sdom: *p* = 0.014; Scom: *p* = 0.047).These findings are in line with those reported in a more recent study by Zoccante et al. [31], which evaluated the impact of an equine-assisted intervention programme involving 86 children with ASD (levels 1 to 3). The authors observed significant improvements in daily living skills and notable gains in socialisation following 20 weekly sessions. These results further support the growing evidence base suggesting that equine-assisted interventions may positively influence autonomy and adaptive functioning in children on the autism spectrum.

Regarding motor skills, there are numerous studies [11,12,13,14,15] that use various measurement tools and show a significant association between the practice of EAI and an improvement in motor skills. However, said association could not be verified in any of the studies that use the VABS, including those conducted by Borgi et al. [27] and by Azjeman et al. [28], where no statistically significant association appears either in the domain or in the subdomains (Dms: *p* = 0.600; Sgr: *p* = 0.892; Sfi: *p* = 0.339), and the same applies to our study (Dms: *p* = 0.363; Sgr: *p* = 0.363; Sfi: *p* = 0.363). Nevertheless, one must be cautious when considering these results due to the restriction of the VABS, which limits observations in the motor skills domain to children under six years old, thereby clearly reducing the sample, especially when our study indeed shows progress—albeit not significant—in the scores (Dms mean pre: 72.50 ± 8.83 ~ mean post: 75.17 ± 7.88; Sgr mean pre: 11.67 ± 4.22 ~ mean post: 12.00 ± 3.74; Sfi mean pre: 12.33 ± 5.04 ~ mean post: 13.00 ± 4.42). This limitation in our study design must be interpreted in light of broader findings in the literature. A recent systematic review by Meera et al. [29] provides compelling evidence of the positive effects of equine-assisted interventions on motor outcomes in autistic children, reporting significant improvements in coordination, strength, balance, posture, and overall motor performance. These results underscore the potential of EAI to promote physical development in this population, even if our current data, constrained by sample size and scale limitations, were not sufficient to yield statistically significant findings.

Finally, it should be noted that no prior studies could be found that examined the effectiveness of EAI in children with ASD as a function of other independent variables. Our study fills this gap by analysing the effectiveness of the intervention as a function of the child’s gender and age, of whether the child received or not complementary therapy or early attention, and of the type of classroom setting, i.e., standard or open. This distinction suggests that the children who are placed in a standard classroom setting and receive early attention achieve better overall results post-intervention (*p* = 0.017/type of classroom; *p* = 0.041/early attention) and greater increases in the test scores (*p* = 0.046/type of classroom; *p* = 0.049/early attention) compared to those who are placed in an open classroom setting and do not receive early attention. With regard to gender, it should be pointed out that although no statistically significant relationship was established, higher averages were observed in boys than in girls, both in terms of post-intervention overall score averages (mean boys: 73.33 ± 18.13 ~ mean girls: 69.25 ± 4.49; *p* = 0.667) and in terms of post-intervention score increases (mean boys: 7.27 ± 9.47 ~ mean girls: 4.25 ± 2.50; *p* = 0.185). Age does not seem to be a significant factor either in the overall scores (*p* = 0.849) or in the increases post-intervention (*p* = 0.280). Finally, the type of complementary therapy session does not seem to be a significant factor either in post-intervention score averages (*p* = 0.762) or in their increases (*p* = 0.736). In addition, paradoxically, the greatest increase in the scores was observed in the group of children that did not receive any type of complementary therapy, followed by those who received psychological support, and finally by those who participated in speech therapy sessions (mean none: 8 ± 12.09 ~ mean psych: 7 ± 3.16 ~ mean speech: 4.33 ± 3.32; t(18): 0.31, *p* = 0.736). The explanation may lie in that the children undergoing speech therapy have more severe ASD, which makes it logical for their post-intervention overall scores and the increases of said scores to be lower.

Nonetheless, several limitations of this study must be acknowledged. Firstly, the relatively small sample size (n = 19), together with the use of a convenience sampling method, limits the statistical power of the analysis and the generalisability of the results. A formal power calculation was not feasible due to the low number of children with autism spectrum disorder (ASD) undergoing equine-assisted intervention (EAI) in our region. Secondly, the absence of randomisation and the use of a single group pre-post design restrict the ability to control for potential confounding variables. Thirdly, all participants were recruited from a single institution—the Tiovivo Foundation—which although it ensured consistency in the implementation of the intervention further constrains the external validity of the findings. The Tiovivo Foundation is the only centre in the area that provides structured EAI programmes with appropriate clinical and educational coordination. Finally, we must acknowledge the potential for co-intervention bias. As participants were receiving various concomitant therapies (e.g., psychological support, speech therapy, and early intervention) throughout the study period, the observed improvements cannot be attributed solely to the EAI. The concurrent support received by the children may have contributed to or interacted with the therapeutic effects of the equine intervention. These factors should be taken into account when interpreting the findings and considering their applicability to broader contexts. Future studies with larger, randomised samples and multicentre collaboration are warranted to confirm and expand upon these results.

## 5. Conclusions

This study suggests preliminary support for the effectiveness of equine-assisted-intervention (EAI) as a complementary therapeutic strategy for children with autism spectrum disorder (ASD). Statistically significant gains were noted in overall adaptive behaviour, particularly in the domains of communication, social skills, and daily living skills following an eight-month EAI programme. While a non-significant improvement in motor skills was noted, consistent with findings from other studies utilising the Vineland Adaptive Behaviour Scale, this observation warrants cautious interpretation due to the inherent limitations of the assessment tool for motor skills in older children.

Furthermore, this research provides initial insight into the influence of educational setting and early intervention on therapeutic outcomes; children attending standard classroom settings and receiving early stimulation appeared to achieve better post-intervention results and greater score increments. These findings point to the potential beneficial impact of EAI programmes and indicate that their efficacy may be enhanced when integrated within a supportive educational environment and alongside early intervention services.

## Figures and Tables

**Figure 1 healthcare-13-02014-f001:**
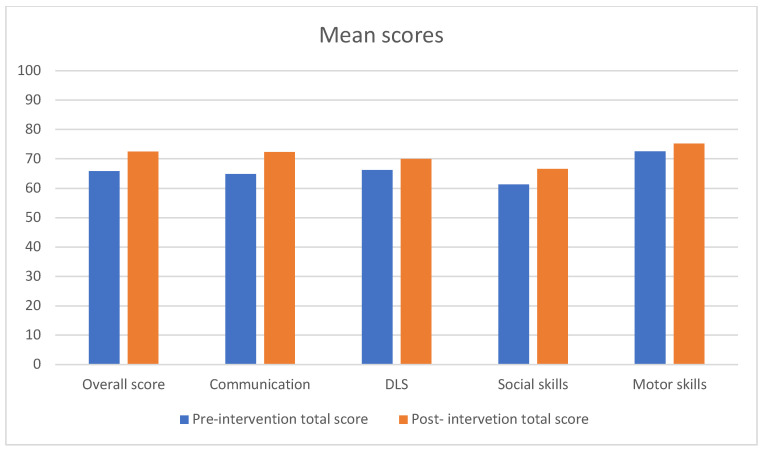
Mean scores pre- and post-EAI in the VABS-II.

**Figure 2 healthcare-13-02014-f002:**
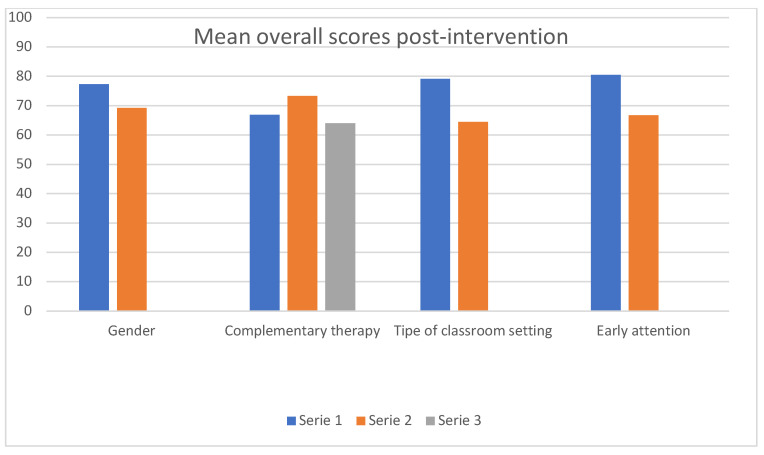
Mean overall scores post-intervention by independent variable.

**Table 1 healthcare-13-02014-t001:** Structure of each EAI session.

Phases	Setting	Duration	Work Tools	Objectives
Initial contact	Common areas	5 min	None	Create child–horse–therapist friendship link Avoid fear towards the horse
Horse preparation/hygiene	Stable	10 min	Harnesses, brushes, hygiene box Rug, surcingle Pictograms	Learning safety standards Learning horse morphology Extrapolation of actions to DLS
Preparation of the child	Harness room	5 min	Helmet, vest, pictographs, sketches	Attention and memory improvement Handling the horse with one’s hand
Riding	Riding school or trail	30 min	Hoops, balls, elastic bands, puppets, stuffed toys	Adaptation of the child to horseback riding and to surroundings, familiarisation with therapist Balance and coordination improvement Improved child–horse–therapist social interaction
Closing of session	Riding school or trail Stable Harness room	10–15 min	Pictographs, blackboard, horse feed (carrots, fruit peel, water)	Establish connection with the horse (touch, caress, hug, give water or food) Improved child–horse–therapist social interaction

**Table 2 healthcare-13-02014-t002:** Mean scores pre- and post-EAI in the VABS-II by domain and subdomain.

Dependent Variables	Pre-Intervention Total Score		Post-Intervention Total Score					
VABS-II	$\bar{x}$	SD	** *p* **	d Cohen	IC 95%	$\bar{x}$	SD	Increase
Overall score	65.84	10.38	0.003	−0.78	[−1.29, −0.27]	72.47	16.21	9.14%
Communication domain	64.84	14.50	0.010	−0.66	[−1.16, −0.16]	72.26	21.93	10.26%
S_receptive_	9.11	2.40	0.007	−0.70	[−1.21, −0.19]	10.58	3.32	13.89%
S_expressive_	7.53	2.98	0.031	−0.54	[−1.04, −0.04]	8.58	4.07	12.23%
S_written_	9.84	4.76	0.043	−0.50	[−0.99, −0.01]	11.00	5.53	10.54%
DLS domain	66.21	11.15	0.0004	−0.99	[−1.52, −0.44]	69.95	12.32	5.34%
S_personal_	9.05	3.51	0.004	−0.75	[−1.26, −0.24]	9.95	4.00	9.04%
S_domestic_	10.05	1.54	0.014	−0.63	[−1.12, −0.12]	10.68	1.56	5.89%
S_community_	8.21	3.02	0.047	−0.49	[−0.98, 0.01]	9.37	2.96	12.37%
Social skills domain	61.26	8.99	0.008	−0.68	[−1.19, −0.18]	66.53	13.79	7.92%
S_interpersonalrelationship_	7.68	2.02	0.008	−0.69	[−1.20, −0.18]	8.68	2.68	11.52%
S_play and leisure time_	7.32	2.49	0.030	−0.54	[−1.04, −0.04]	8.47	3.53	13.57%
S_copingskills_	9.05	1.31	0.150	−0.34	[−0.82, 0.13]	9.63	2.31	6.02%
Motor skills domain	72.50	8.83	0.363	−0.41	[−1.53, 0.72]	75.17	7.88	3.55%
S_gross motor_	11.67	4.22	0.363	−0.41	[−1.53, 0.72]	12.00	3.74	2.75%
S_fine motor_	12.33	5.04	0.363	−0.41	[−1.53, 0.72]	13.00	4.42	5.15%

Abbreviations: EAI, Equine-Assisted Intervention; VABS-II, Vineland Adaptive Behavior Scale-II; DLS, Daily Living Skills; IC, Confidence Interval. The values represent the mean ± standard deviation. The *p*-value was calculated using a paired Student’s *t*-test. The statistical significance criterion was set at *p* < 0.05.

**Table 3 healthcare-13-02014-t003:** Mean overall scores post-intervention and mean score increases from pre- to post-intervention in the VABS-II, by independent variable subgroups.

Independent Variables	Mean Overall Scores Post-Intervention					Mean Difference in Overall Scores from Pre- to Post-Intervention				
	$\bar{x}$	SD	*p*	d Cohen	IC 95%	$\bar{x}$	SD	*p*	d Cohen	IC 95%
Gender			0.667	0.24	[−0.75, 1.23]			0.544	0.34	[−0.65, 1.33]
Boys (n = 15)	73.33	18.13				7.27	9.47			
Girls (n = 4)	69.25	4.92				4.25	2.50			
Complementary therapy			0.762	0.12	[−0.75, 0.99]			0.736	0.14	[−0.72, 1.00]
None (n = 9)	66.89	10.84				8.00	12.09			
Psychological support (n = 4)	73.25	8.95				7.00	3.16			
Speech therapy (n = 6)	64.00	12.97				4.33	3.32			
Type of classroom setting			0.017	1.21	[0.34, 2.08]			0.046	1.04	[0.08, 1.99]
Standard (n = 11)	79.09	12.21				10.64	11.49			
Open (n = 8)	64.50	11.37				2.63	2.61			
Early attention			0.041	1.05	[0.08, 2.01]			0.049	1.16	[0.08, 2.24]
Yes (n = 7)	80.43	15.98				13.43	12.43			
No (n = 12)	66.75	10.98				1.83	2.29			
Age (n = 19)	8.37	C. Pearson	0.849			8.37	C. Pearson	0.280		
VABS-II total score	72.47	0.047				6.63	−0.261			

Abbreviations: VABS-II, Vineland Adaptive Behavior Scale-II; IC, Confidence Interval. The values represent the mean ± standard deviation. Subgroup comparisons and correlational analyses were performed using independent *t*-tests, a one-way ANOVA, and Pearson’s correlation coefficient, respectively. The statistical significance criterion was set at *p* < 0.05.

## Data Availability

The data presented in this study are available upon reasonable request from the corresponding author. An anonymised dataset, along with the SPSS syntax used for the analyses, can be provided to interested researchers subject to a signed data sharing agreement. This measure is in place to ensure participant anonymity and to comply with the ethical approval of the study.

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
