# Peer review of "Effectiveness of Equine-Assisted Intervention as a Therapeutic Strategy for Improving Adaptive Behaviour in Children with Autism Spectrum Disorder"

_healthcare, 2025, doi:10.3390/healthcare13162014_

Round 1

Reviewer 1 Report

Comments and Suggestions for Authors

1.    The main problem of this study is the number of subjects; the number of subjects studied is relatively small. How was the sample size estimated? The Methods section should include A formal power calculation to justify the feasibility of detecting clinically meaningful changes with the current cohort. Additionally, the generalizability of findings is constrained by the single-center recruitment (Tiovivo Foundation) and lack of randomization. These limitations should be expanded upon in the discussion..
2.    The other problem of this study is the absence of control, which is a significant confounder. The self-controlled design (pre/post-intervention) cannot distinguish EAI effects from natural development, concurrent therapies, or placebo effects. This confounder gives participants ongoing access to conventional therapies (e.g., speech therapy, psychological support). Please include a matched control group (e.g., waitlist or alternative intervention) to isolate EAI-specific effects. Alternatively, you can utilize propensity scoring to adjust for confounding variables.
3. The interpretation of motor skills outcomes requires clarification. The VABS-II motor domain was assessed in only six participants due to age restrictions (applicable only to children under 6 years). While the authors note the non-significant trend toward improvement, the small subsample undermines the validity of any conclusion in this domain. It is recommended that this section be reframed as exploratory, that statistical testing for motor skills be omitted, or that the analysis be stated as underpowered.
4. No data on ASD severity (e.g., ADOS scores), symptom profiles, or therapy dosage/duration
5. In Table 2, the pre-intervention standard deviation for the Communication domain is listed as 14.50 in the table but reported as 15.50 in the text (line 29). This discrepancy must be corrected. 
6. The discussion comparing results with Azjenman et al. [28] contains a typographical error: "Dcom: p=0.027; this study: Dcom: p=0.008" should refer to the social skills domain (Dsoc), not Dcom. 
7. The term "open classroom" is used without definition. 
8. Approval by an "animal study protocol" (Line 327) for human subjects research requires clarification.

Reviewer 2 Report

Comments and Suggestions for Authors

Manuscript: healthcare-3793943

Title: “Effectiveness of Equine Assisted Intervention as a Therapeutic Strategy for Improving Adaptive Behavior in Children with Autism Spectrum Disorder”

Date: 26 July 2025

Dear Authors,

Your article presents a study on equine‑assisted intervention for children with autism spectrum disorder (ASD). The topic is of considerable interest and, with appropriate improvements, can provide valuable evidence to the literature on animal‑assisted therapies. After a careful reading of the main manuscript and the supplementary material, I believe the paper requires a thorough revision. Below you will find my comments organised by section, with concrete indications of the changes needed.

1. Theoretical framework and references: Please update the literature review with studies from 2024–2025 on equine‑assisted interventions and, more broadly, animal‑assisted therapies in ASD. In the Introduction, underscore the specific gap your study addresses (e.g., duration, cultural context, participant age, etc.).

2. Objectives and study framing: Reformulate the study’s purpose to make clear that this is an exploratory pilot of feasibility and possible effect, not a definitive efficacy trial, as no control group was included. A pre–post design without a control group does not allow changes to be attributed exclusively to the intervention; threats such as maturation, history, and concomitant treatments remain. Explicitly acknowledging this exploratory frame strengthens scientific integrity and avoids over‑interpreting results.

Accordingly, moderate causal claims; expressions such as “demonstrates efficacy” or “proves that” should be replaced with “suggests,” “provides preliminary indications,” etc.

4. Methodology and statistical analysis: Please add several elements that are basic requirements for transparency and methodological rigour:

a) Power calculation or post‑hoc analysis: specify the minimally clinically important effect size, Type I error (α), intended power (1–β), and explain whether the sample of 19 participants is sufficient to detect it.

b) Effect sizes and confidence intervals: alongside each p value, report the appropriate effect size (e.g., Cohen’s d or r) with its 95% CI.

c) Adjustment for multiple comparisons: justify analysing multiple VABS‑II domains and subdomains without correction, or apply a method (Bonferroni, Holm, FDR).

d) Confounders: describe other therapies or educational supports received over the eight months and explain how they might have influenced the outcomes.

e) Motor domain: clarify why it was assessed only in six children under six years of age and discuss the implications for comparability with the overall results.

4. Results: I recommend the following:

a) Restructure the Results section to separate the objective description of data from any interpretation or discussion.

b) Present absolute and relative (percentage) changes in VABS‑II scores, not just overall means.

c) Add simple figures/graphs (e.g., bar charts with CIs) to illustrate the magnitude of improvements.

d) Standardise decimal notation (dot or comma, not both) and statistical nomenclature across all tables.

5. Quality of english: Please seek a professional language edit. It is necessary to shorten very long sentences, remove literal translations from Spanish, maintain consistent verb tense, and use standard terminology (e.g., “equine‑assisted intervention” rather than “hipicoterapia”).

I believe that, once the above points are addressed, the manuscript could contribute to the preliminary evidence on equine‑assisted interventions in paediatric ASD.

Kind regards,

External Reviewer — Healthcare (MDPI)

Round 2

Reviewer 1 Report

Comments and Suggestions for Authors

The authors responded to all issues 

Reviewer 2 Report

Comments and Suggestions for Authors

Manuscript: healthcare-3793943

Title: “Effectiveness of Equine Assisted Intervention as a Therapeutic Strategy for Improving Adaptive Behavior in Children with Autism Spectrum Disorder”

Date: August 8, 2025

Dear Authors,

Thank you for sending the revised version. The work is evident: you have reframed the study as an exploratory pilot, incorporated effect sizes with confidence intervals, justified the handling of multiple comparisons, and updated the bibliography. I also appreciate the inclusion of figures and the clarification on the application of the VABS-II motor domain. With these advances, the manuscript is close to being publishable. Below are the final adjustments—all feasible without collecting new data—that, in my opinion, will make the article ready for publication.

First, I would appreciate a little more detail about concomitant therapies and other support received during the intervention period. If the information is available, please indicate at least the approximate frequency (e.g., weekly speech therapy sessions, biweekly early intervention, etc.). If it is not possible to specify the “dose,” simply describe the presence/absence by type of support and add a paragraph in Limitations on the meaning of possible bias (e.g., co-intervention may have contributed to part of the observed improvement). If you have already made any exploratory comparisons by subgroups, a brief mention in Results (without overinterpreting) would add transparency.

Second, an editorial polish of numbers and tables. Please standardize the decimal notation (period or comma, always the same in text, tables, and figures) and correct the small errors that have still slipped into labels and figure captions. In Tables 2–3, add footnotes that make each table “self-contained”: definition of all abbreviations (VABS-II, DLS, IC/CI, etc.), statistical test used (e.g., paired t-test), significance criterion, and the exact name of the reported effect size. This prevents the reader from having to go to the end of the manuscript to decipher the terms and greatly improves readability.

Third, a final review of the English. The overall writing has improved, but some non-idiomatic phrases and typographical errors in titles and figure captions remain. A professional edit (or at least a critical reading by a native speaker) will resolve in a day what might otherwise hold up production.

Finally, I suggest adjusting the order of the final sections to that recommended by Healthcare (IRB/Consent/Data Availability/Conflicts/Funding/Acknowledgments). It is a minor detail, but it facilitates the layout work and avoids back-and-forth in the final phase. If you can also expand the Data Availability Statement by one line (indicating the format and conditions of access to an anonymized dataset and, if feasible, to the SPSS syntax), you would be fully aligned with good transparency practices.

That's all from me. Congratulations on the progress between versions: with these minor adjustments, I believe the manuscript will be ready for acceptance as a pilot study. I remain at your disposal if you need any specific wording to incorporate into the sections mentioned.

Kind regards,

External Reviewer — Healthcare (MDPI)

Comments on the Quality of English Language

The overall writing has improved, but some non-idiomatic phrases and typographical errors in titles and figure captions remain. A professional edit (or at least a critical reading by a native speaker) will resolve in a day what might otherwise hold up production.
